# Melatonin in Endometriosis: Mechanistic Understanding and Clinical Insight

**DOI:** 10.3390/nu14194087

**Published:** 2022-10-01

**Authors:** Yiran Li, Sze-Wan Hung, Ruizhe Zhang, Gene Chi-Wai Man, Tao Zhang, Jacqueline Pui-Wah Chung, Lanlan Fang, Chi-Chiu Wang

**Affiliations:** 1Department of Obstetrics & Gynaecology, The Chinese University of Hong Kong, Hong Kong 999077, China; 2Center for Reproductive Medicine, Henan Key Laboratory of Reproduction and Genetics, The First Affiliated Hospital of Zhengzhou University, Zhengzhou 450052, China; 3Laboratory of Reproduction and Development, Li Ka Shing Institute of Health Sciences, The Chinese University of Hong Kong, Hong Kong 999077, China; 4School of Biomedical Sciences, The Chinese University of Hong Kong, Hong Kong 999077, China; 5Chinese University of Hong Kong-Sichuan University Joint Laboratory in Reproductive Medicine, The Chinese University of Hong Kong, Hong Kong 999077, China

**Keywords:** melatonin, endometriosis, anti-inflammatory, inflammatory disease, clinical trials, in vitro studies, animal studies

## Abstract

Endometriosis is defined as the development of endometrial glands and stroma outside the uterine cavity. Pathophysiology of this disease includes abnormal hormone profiles, cell survival, migration, invasion, angiogenesis, oxidative stress, immunology, and inflammation. Melatonin is a neuroendocrine hormone that is synthesized and released primarily at night from the mammalian pineal gland. Increasing evidence has revealed that melatonin can be synthesized and secreted from multiple extra-pineal tissues where it regulates immune response, inflammation, and angiogenesis locally. Melatonin receptors are expressed in the uterus, and the therapeutic effects of melatonin on endometriosis and other reproductive disorders have been reported. In this review, key information related to the metabolism of melatonin and its biological effects is summarized. Furthermore, the latest in vitro and in vivo findings are highlighted to evaluate the pleiotropic functions of melatonin, as well as to summarize its physiological and pathological effects and treatment potential in endometriosis. Moreover, the pharmacological and therapeutic benefits derived from the administration of exogenous melatonin on reproductive system-related disease are discussed to support the potential of melatonin supplements toward the development of endometriosis. More clinical trials are needed to confirm its therapeutic effects and safety.

## 1. Introduction

Melatonin (N-acetyl-5-methoxytryptamine) is a neurohormone synthesized and released at night from the pineal gland. This hormone has been shown to regulate the circadian rhythm of the body, pubertal development, and seasonal adaptation [1,2]. It has also been reported to act directly on the hypothalamic–pituitary–gonad (HPG) axis to regulate reproduction [3,4]. For many years, melatonin has only been considered as a hormone to be secreted exclusively by the pineal gland. In recent years, it has been shown that this hormone can be produced by many extra-pineal tissues to exert its biological functions locally [5]. As a pleiotropic molecule, melatonin can interact with female reproductive organs, including the ovaries, uterus, mammary glands, and placenta, to regulate many physiological and pathological events [6,7]. Several studies have shown the capabilities of melatonin to improve fertilization in sub-fertile patients [8]. The beneficial effects of melatonin in reproduction can be attributed to its well-characterized antioxidative effect and other biological functions, such as the regulation of steroid hormone production, anti-cell proliferation, pro-apoptosis, anti-cell adhesion, anti-invasion, anti-epithelial-mesenchymal transition (EMT), anti-angiogenic activity, and regulation of immune modulation as well as neurotrophic effects [9].

Endometriosis is a chronic disease that can be defined as the development of endometrial glands and stroma on the uterine cavity [10]. It is estimated that endometriosis affects 5 to 10% of women between 25 and 35 years old [10]. Endometriosis is an estrogen-dependent inflammatory disease that affects women at reproductive stage and leads to sterility. Typical symptoms of endometriosis include chronic pelvic pain, dysmenorrhea, and dyspareunia.

The pathogenesis of endometriosis is not yet fully understood. Despite several proposed theories on the etiology of endometriosis, none of these models can fully explain the pathogenesis of this disease. Regardless of this, it is commonly theorized to arise from the endometrial fragments escaping into the peritoneal cavity through the process of retrograde menstruation. Meanwhile, abnormal endocrine, immunology, pro-inflammatory, and proangiogenic pathways may contribute to the development of endometriosis [11,12]. Additionally, many abnormal cellular functions, including cell proliferation and migration/invasion, have been shown to induce the process [13]. Misinterpretation of endometriosis-induced pain, such as menstruation-associated abdominal cramps, can often delay the diagnosis of endometriosis by up to 8 to 10 years [14]. Treatment of endometriosis is mainly used to alleviate the symptoms and increase fertility rates [15]. Hormonal compounds and symptomatic treatments are the major medications to treat endometriosis, including non-steroidal anti-inflammatory drugs, gonadotropin-releasing hormone agonists, progestins, and oral contraceptive pills [16]. As these medications bring side effects by creating a hypoestrogenic state to suppress ovarian functions, women of childbearing age should instead only use pain relief [15]. As a consequence, it is highly crucial to develop potential new treatments that do not interfere with the reproductive functions in women suffering from endometriosis. As the occurrence of endometriosis cannot be explained by any single pathophysiological event, therapeutic drugs should be multi-targeted to resolve the disease complexity [17]. Regarding this, melatonin could be a potential candidate as an ideal medication for the treatment of endometriosis. With melatonin demonstrating a potential therapeutic effect against endometriosis in an early clinical study, its therapeutic targets and molecular mechanism are worth exploring further [18].

Considering that several melatonin-involved pathways might be relevant to the pathophysiology of endometriosis, there is currently no review dedicated to melatonin as a potential treatment of endometriosis. Hence, a more comprehensive evaluation of the effects of melatonin in treating endometriosis is worth deliberating. This review aims to explain the biology and molecular mechanism of melatonin and emphasizes its potential clinical applications for treating endometriosis.

## 2. The Biology of Melatonin

### 2.1. Synthesis

The classic melatonin biosynthetic pathway includes five enzymatic steps in animals. In the first step, tryptophan is hydroxylated to 5-hydroxytryptophan by tryptophan-5-hydroxylase (TPH); consequently, it is decarboxylated to serotonin (5-hydroxytryptamine) by the aromatic amino acid decarboxylase (AADC). In the two final steps, it is proposed that serotonin is acetylated by aralkylamine N-acetyltransferase (AANAT) to N-acetylserotonin. Ultimately, this works as an intermediate to form N-acetyl-5-methoxytryptamine, also known as melatonin, exclusively in the pineal gland in animals [19]. After synthesis from the pineal gland, melatonin passes to the hypothalamic suprachiasmatic nuclei (SCN) via the polysynaptic pathway. Additionally, in recent year, multiple studies have also shown that melatonin could be produced in several extra-pineal organs, including the retina, skin, gastrointestinal tract, and bone marrow, and act locally through autocrine and/or paracrine manners [20], while for the female reproductive system, melatonin was found to be secreted by the uterus, ovaries, and placenta [21].

### 2.2. Metabolism

The circadian melatonin cycle is at maximum at night and minima during the day. The plasma half-life of melatonin is relatively short, around 20 to 50 minutes [22]. Melatonin is highly soluble in both water and lipids. Therefore, it easily diffuses through the cell membrane and the blood–brain barrier [23]. In humans, melatonin is mainly metabolized to 6-hydroxymelatonin (6-HMEL) that serves as a surrogate of circulatory melatonin levels by the liver enzyme cytochrome P450 1A2 (CYP1A2), which is then subsequently conjugated with sulfate and is excreted in the urine [24]. For exogenous melatonin, the peak plasma/serum concentration (Tmax) is achieved at approximately 50 min after oral administration of melatonin in an immediate-release dosage form. In accordance, the elimination half-life (T1/2) of melatonin for both oral and intravenous administration was 45 min [25].

### 2.3. Receptors

In humans, there have been two main isoforms of the membrane-bound melatonin receptors (MTs), namely MT1 and MT2, identified in the plasma and peripheral tissues [22]. Both of them belong to the G protein-coupled receptors (GPCRs) family. The key biological effects of melatonin are exerted by activating through MT1 and MT2 in humans [22]. After the two receptors couple to Gαi/o proteins, intracellular levels of the second messenger cyclic adenosine monophosphate (cAMP) are consequently decreased, which is the most common signaling pathway activated by melatonin [26], such that MT1 and MT2 receptors have a high density of expressions in the neuroendocrine system, such as the hypothalamic SCN, pituitary pars tuberalis (PT), and gonadotroph cells [27,28]. These receptors are also expressed in several peripheral organs, including skin, vasculature, gastrointestinal tract, and kidney [29,30,31,32]. In the reproductive system, melatonin receptors can be detected in the placenta, uterus, and endometrium [33,34,35]. These biological effects are mediated by the interaction of melatonin with MT1/MT2 for activating other molecular pathways. For instance, melatonin may act as a ligand for the family of orphan retinoid-linked nuclear receptors (RZR/ROR), including RZRα, RORα, RORα2, and RZRβ [36].

### 2.4. Biological Effects

Besides being known to regulate the circadian rhythm, new evidence indicates that melatonin can act as a regulator in the endocrine, cardiovascular, immune, and neutral systems [37,38,39,40]. Melatonin exerts these effects by working as a potent antioxidant that can directly scavenge free radicals, oxygen-centered radicals, and other toxic hydroxyl radicals [41,42]. Owing to its potent antioxidant properties, melatonin can promote apoptosis with anti-inflammatory and anti-angiogenesis capabilities toward various carcinomas [43,44,45]. Likewise, melatonin has been shown to play a key role in reproduction and women’s health. For instance, studies have shown a significant and positive impact on the maturation of the oocyte and subsequent embryonic development [46]. In addition, patients who were given melatonin during in vitro fertilization and embryo transfer (IVF-ET) are likely to increase the success rate of the IVF-ET treatment [47]. Regarding endometriosis, women who worked during night shifts were reported to have decreased pineal gland melatonin production and an increased risk of endometriosis, hinting at a possible correlation between melatonin and endometriosis [48,49].

## 3. Potential Therapeutic Mechanisms of Melatonin in Endometriosis

In studies using a model of endometriosis, melatonin was shown to prevent and treat endometriosis by reducing the size and weight of the endometriotic lesions [50,51,52], while in women with endometriosis, MT2 protein expression was significantly decreased in peritoneal lesions compared to eutopic endometrium [53]. A randomized phase II double-blinded, placebo-controlled clinical trial demonstrated that melatonin could alleviate pain in endometriosis [18]. Details are summarized below.

Current evidence suggests that melatonin supplements are safe for short-term use [54]. Even when taken at extreme doses, only mild adverse effects such as dizziness, sleepiness, nausea, and headache have been reported [55,56]. No studies have indicated that exogenous melatonin would induce any serious adverse effects. In addition, clinical studies have proven the safety profile of exogenous melatonin to be used as a supplement, particularly in obstetrics and reproductive-related conditions, such as for fertility [57]. These studies showed the potential of melatonin as a therapeutic drug for endometriosis without any negative effect on the reproductive system. Similarly, randomized clinical studies indicate that long-term melatonin treatment causes only mild adverse effects comparable to a placebo. Long-term safety of melatonin in children and adolescents, however, requires further investigation. The schematic diagram of the potential therapeutic effects of melatonin on endometriosis is shown in Figure 1. The studies related to the determined effects of melatonin on endometriosis are summarized in Table 1. In addition, the effects of exogenous melatonin in the rat model of endometriosis are compared in Figure 2.

**Table 1 nutrients-14-04087-t001:** Experimental studies of melatonin for endometriosis.

Studies	Study Type	Study Sample Specie	Models/Methods	Dose	Route	Duration	Results	Mechanism
Guney 2008 [58]	Animal in vivo	rats	Endometrium fragments sutured to the abdominal wall	10 mg/kg daily	ip	4 weeks	Decreased lesion sizes and weight	Increased SOD, CAT Decreased MDA, COX-2
Paul 2008 [59]	Animal in vivo	mice	Intraperitoneal transplantation	48 mg/kg daily	ip	10 or 20 days	/	Increased TIMP-1 Decreased MMP-9
Paul 2010 [60]	Animal in vivo	mice	Intraperitoneal transplantation	48 mg/kg daily	ip	10 or 20 days	/	Increased TIMP-3 Decreased MMP3
Yildirim 2010 [61]	Animal in vivo	rats	Endometrium fragments sutured to the abdominal wall	10 mg/kg daily	ip	2 weeks	Decreased lesion sizes Decreased histopathologic scores	Increased SOD, CAT
Koc 2010 [62]	Animal in vivo	rats	Endometrium fragments sutured to the abdominal wall	10 mg/kg daily	ip	4 weeks	Decreased histopathologic score	Increased SOD, CAT Decreased MDA
Kocadal 2013 [63]	Animal in vivo	rats	Intraperitoneal transplantation	20 mg/kg daily	im and ip	2 weeks	Decreased lesion sizes Decreased histopathologic score	/
Yilmaz 2015 [51]	Animal in vivo	rats	Endometrium fragments sutured to the abdominal wall	10 mg/kg daily	ip	4 weeks	Decreased lesion weight Decreased histopathologic scores	Increased SOD, MDA, TIMP-2 Decreased VEGF, MMP-9
Cetinkaya 2015 [50]	Animal in vivo	rats	Endometrium fragments sutured to the abdominal wall	10 or 20 mg/kg daily	im and ip	2 weeks	Decreased lesion sizes in both treatment groups without significant difference	/
Shasha 2018 [64]	Cell in vitro	human	Endometriotic eutopic epithelial cell culture	1 mM	culture medium	72 h	/	Increased Numb, E-cadherin Decreased migration, invasion and Notch1, Vimentin, Slug, Snail

ip, Intraperitoneal injection; im, Intramuscular injection; SOD, Superoxide dismutase; CAT, Catalase; MDA, Malondialdehyde; COX-2, Cyclooxygenase 2; TIMP, Tissue inhibitors of metalloprotease; MMP, Matrix metalloproteinase; VEGF, Vascular endothelial growth factor.

**Figure 2 nutrients-14-04087-f002:**
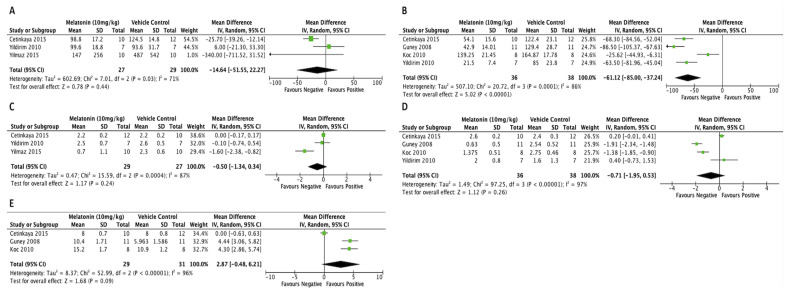
Forest plots of effectiveness of melatonin treatment compared with control in endometriosis. Forest plot of included studies comparing the effect of exogenous melatonin (10 mg/kg) in a rat model of endometriosis. (**A**) Lesion size at 2 weeks [50,51,61]. (**B**) Lesion size at 4 weeks [50,58,61,62]. (**C**) Histopathological score after 2 weeks [50,51,61]. (**D**) Histopathological score after 4 weeks [50,58,61,62]. (**E**) Level of superoxide dismutase (SOD) after 2 weeks [50,58,62].

### 3.1. Anti-Oxidation

As endometriosis patients are prominently known to have an upregulated level of reactive oxygen species (ROS), the insufficient antioxidants would prompt a deleterious effect from the imbalance between ROS and antioxidants [65]. The elevated ROS in endometriosis can lead to cellular damage and other undesirable side effects, including a systemic inflammatory reaction and endometriosis-related infertility [66]. With melatonin being a free-radical scavenger and a potent stimulator of antioxidant enzymes, it has been proposed as a potential therapeutic agent to prevent the onset of endometriosis by enhancing the antioxidant levels. Based on an animal study, melatonin was shown to downregulate cyclooxygenase 2 (COX-2) and malondialdehyde (MDA) protein levels and upregulate SOD and catalase (CAT) in the endometriotic lesions [58], while in another in vivo study, melatonin also demonstrated upregulating levels of antioxidants, such as SOD and CAT [52]. Likewise, activities of lipid peroxidation (LP) were induced, and antioxidants, including SOD and CAT, were reduced in an endometriosis rat model after undergoing pinealectomy [62]. Interestingly, these effects were effectively reversed by the introduction of exogenous melatonin [62]. Similarly, the MDA levels and histopathological scores of endometriosis were significantly reduced, with no alteration in SOD levels, in the human xenograft model of endometriosis when treated with melatonin for 4 weeks [67]. Another study also revealed similar results in demonstrating the capability of melatonin to upregulate SOD levels and decreased histopathological scores, but with a discrepancy in the increased MAD level [51].

In general, melatonin has been shown to elicit a tendency to prevent oxidative stress in organs requiring high oxygen consumption (e.g., the brain) and those colonized with certain microbes to create a hostile environment, such as in the gut and skin [68,69]. Besides these, the beneficial role of melatonin as a potent antioxidant has also been demonstrated in the human reproductive system. As melatonin can reduce oxidative stress in follicles, it functioned by preventing the free radicals generated from causing harm on the oocytes [47]. In addition, melatonin can improve the uterine microenvironment by promoting the expression of antioxidant enzymes, such as SOD and CAT [70]. Moreover, these findings clearly illustrate that melatonin can potentially act as an antioxidant to treat endometriosis through the upregulation of antioxidant enzymes and downregulation of pro-oxidant enzymes. The studies related to the determined effects of melatonin on endometriosis are summarized in Table 1.

### 3.2. Regulation of Steroid Hormone Production and Function

Endometriosis is a disease of hormone imbalance, particularly referred to as an estrogen-dependent and progesterone (P4)-resistance disorder [10]. In contrast to normal endometrium, endometriotic tissues have a higher capability to synthesize E2 locally, due to the high expression of two of the key enzymes for estrogen biosynthesis: aromatase (CYP19A1) and steroidogenic acute regulatory protein (StAR) [71]. As the accumulated E2 can induce cell proliferation and inflammation, this would further stimulate the advancement and progression of endometriosis by promoting lesion growth and undermining the endometrial sensitivity [12,72]. In normal endometrium, the mediators of estrogenic action can be through the two receptors estrogen receptor alpha (ERα) (ESR1) and estrogen receptor beta (ERβ) (ESR2) [73]. Their regulation in endometriosis remains uncertain, such that one study suggested that patients with endometriosis would have downregulation of ERα and upregulation of Erβ [74]. On the contrary, another study showed that ERα and ERβ expression levels were both increased in ectopic tissue in comparison with the normal and eutopic endometrium [74]. Although there were mixed results, both studies supported that the ESR2/ESR1 ratio was increased in the endometriotic lesions. The upregulated estrogen activities would therefore cause progesterone resistance [75]. In addition, patients with endometriosis were reported to be unable to manifest P4 to activate its receptor (PGR) or the transcription of its target gene [76]. As a consequence, PGR-B in mRNA and protein concentrations were significantly downregulated in both endometriotic lesions and endometrium [77,78,79]. In this regard, current studies have unveiled the regulatory effects of melatonin on P4 and E2 that would take place in tissue-specific manners and through a series of ongoing mechanisms [80].

#### 3.2.1. Estrogen

Significant research has confirmed the tendency of melatonin in suppressing E2 production. A study conducted on hamster ovary showed that melatonin inhibited E2 synthesis from the preovulatory follicle via the MAPK pathway following treatment with human chorionic gonadotropin (hCG), a hormone to induce the corpus luteum to secrete E2 [81]. Long-term melatonin treatment in rats has been shown to significantly reduce 17 beta-estradiol levels in plasma, ERα expression levels in the ovary, and ERβ levels in the uterine tissue [82]. In a clinical trial, daily administration of 300 mg of melatonin for 4 months resulted in a significant reduction in estradiol level [83]. In preclinical studies on hormone-dependent breast cancer, melatonin was found to lower serum E2 levels, ESR expressions, and tumor growth in mice models, as well as reduce cell viability through the estrogen response pathway in a human breast cancer cell line [84,85,86]. On the other hand, melatonin was noted to stimulate the production of E2 through the activation of the PKA-CREB signaling pathway in human granulosa cells [87,88]. Aromatase, which is encoded by CYP19A1, is the core enzyme to catalyze the androgen biosynthesis to form E2 [89]. In human MCF-7 breast cancer cells, aromatase expression was suppressed by the bounding of melatonin to MT1 for inhibiting downstream functions, such as cell growth, via the cAMP pathway [90,91]. Likewise, melatonin also can reduce E2 biosynthesis and blocked the growth of human umbilical vein endothelial cells (HUVECs) and rat glioma cells via the suppression of the aromatase level [92,93]. Interestingly, melatonin had a stimulating effect on the aromatase expression level in human granulosa cells [88]. The underlying cause of differential effects of melatonin could be explained by CYP19A1 being driven by different promoters at the upstream of the first exons in various tissues, which ultimately leads to the respective outcomes in different tissues [94,95]. Pinealectomy in rats has been reported to increase E2 synthesis and interfere with LH and FSH releases to result in anovulation [96]. Treatment with melatonin inhibited the proliferation of endometrial epithelial cells in the presence of E2 at 48 h after exposure [53]. In a rat endometriosis model, treatment with melatonin resulted in greater regression of endometriosis foci and reduced recurrence rate when compared to those treated with an aromatase inhibitor, letrozole [52]. Moreover, these studies illustrated that melatonin would not only regulate the E2 response pathway but would contribute to suppressing the E2 biosynthesis via aromatase activity.

#### 3.2.2. Progesterone

To date, there is a great deal of related evidence suggesting that melatonin may play a role in regulating P4 resistance. In human granulosa cells, multiple findings showed that melatonin increased the production of P4 [97,98]. On the contrary, a study conducted on patients with uterine cancer showed that melatonin inhibited the production of P4 in human granulosa cells [99]. Meanwhile, some studies showed that melatonin had no direct effect on the production of P4 in human granulosa cells [100,101]. P4 resistance was a consequence of PGR dysregulation, as well as due to the reduction of StAR, a key enzyme in regulating the synthesis of P4. Melatonin enhanced PGR expression in mouse cumulus cells and induced the expression of StAR in human granulosa and bovine theca cells via the activation of the PI3K/AKT signaling pathway [98,102]. A similar upregulation effect was also observed in mammalian Leydig cells [103]. Other contradictory findings suggested that melatonin reduced PGR expression in the MCF-7 human breast cancer cell line [104,105]. P4 is essential for the development of endometrial pinopodes and microvilli. With the levels of P4 being reduced after pinealectomy, it further contributes to the outcome of a reduced number of implantations, indicating the potential role of melatonin [106]. Nevertheless, further studies on the function of melatonin on P4 regulation would still be warranted on hormonal balance.

### 3.3. Anti-Proliferation and Pro-Apoptosis

The endometrium maintains complex controls on proliferation and apoptosis as part of repetitive menstrual cycles that prepare the endometrium for the window of implantation and pregnancy. However, the occurrence of endometriosis would disrupt the balance between cell proliferation and apoptosis [10]. Melatonin was shown to regulate several pathways associated with proliferation and apoptosis in normal and cancer cells [107]. These pathways mainly mediate through nuclear factor-kappaB (NF-κB), PI3K/Akt, and cyclin-dependent kinases (CDKs) to promote cell proliferation [108], such that the regulatory activity of melatonin can be demonstrated by its capability to inhibit NF-κB activation in human lung cancer cell lines and liver cancer cell lines [109,110]. Likewise, melatonin (or in combination with vitamin D3) was found to inhibit proliferation via the PI3K/Akt pathway in a human breast cancer cell line [111,112]. Additionally, melatonin was shown to inhibit the RNA expressions of cyclins and CDKs to suppress the progression of the G1-phase in dopaminergic neuroblastoma cells in rats [113]. In addition, melatonin can inhibit the RNA expression level of cyclin D1 in human breast cancer cells and the RNA and protein expression levels of cyclin D1, cyclin B1, CDK1, and CDK4 in osteosarcoma cells [114,115]. Although there is currently no literature documenting the capability to downregulate proliferation via the above-mentioned pathways in endometriosis, several studies have shown that melatonin significantly suppressed eutopic endometrial lesions in endometriosis mouse models [52,58,63]. Additionally, melatonin could also inhibit the cell proliferation induced by estrogen in human endometrial epithelial cells [53]. However, melatonin can facilitate apoptosis through the modulation of various apoptotic mediators including mitochondria, Bax, Bcl-2, endogenous ROS, and apoptosis receptors [116]. In a study using a xenograft of endometriosis, it showed that the treatment of melatonin can enhance apoptotic cells while reducing the expression of Bcl-2 via the upregulation of Bax expression and promoting caspase 9 activation [60]. Taken together, these studies can indicate the potential of melatonin to possibly restore the balance of the endometrium on proliferation and apoptosis disrupted by the occurrence of endometriosis.

### 3.4. Anti-Adhesion and Anti-Invasion

Ectopic endometriosis cells have an advanced potential to invade immunosurveillance and attach to the lymph nodes and the abdominal cavity [117]. Viable glandular and stromal endometrial cells attach to the abdominal cavity through interacting with cell surface receptors, including integrins and membrane adhesion molecules in the extracellular matrix (ECM), such as fibronectin and laminin [117,118]. The mRNA and protein levels of collagen I and fibronectin were found to be inhibited by melatonin in a hindlimb ischemia mouse model [119]. Matrix metalloproteinases (MMPs) are a large family that can remodel ECM and regulate adhesion and invasion of cells [120]. MMP proteins were highly expressed in endometriotic tissues when compared to normal endometrium [121,122,123]. As the tissue inhibitors of metalloproteases (TIMPs) are endogenous inhibitors of MMPs, a significant reduction in the protein expressions of TIMP-1 and TIMP-2 in endometriotic diseased tissue was reported [124,125,126]. In another study, it was shown that eutopic endometrium and ectopic lesions of women with endometriosis had lower levels of TIMP-3 mRNA expression, while the MMP-9 mRNA level was solely enhanced in ectopic lesions [121]. Other studies also showed that the expression levels of MMP2, MMP3, and MMP9 in ectopic endometrium and endometriosis cells were amplified [127,128]. In accordance, melatonin was found to regulate MMP gene expression and activity in many diseases, such that melatonin could protect against endometriosis by regulating TIMP-1/MMP9 and TIMP3/MMP3 in the peritoneal endometriosis mice model on degrading ECM and regressing the growth of the endometriotic lesion [59,60]. Similarly, melatonin was found to increase TIMP-2 expression levels in the endometriosis rat model [51]. On the other hand, melatonin was shown to downregulate MMP2 and MMP9 expression levels in the gastric mucosa of rats, human gastric cancer cell lines, and human breast cancer cell lines [129,130,131]. It was also found to reduce the concentration of MMP9 in rat retina, human ovarian cells, and breast cancer cells [132,133,134]. The downregulation in mRNA and protein levels of MMP2 and MMP9 can induce high glucose and IL-1β levels in human retinal endothelial cells and retinal pigment epithelial cells [135]. In addition, mice that were kept under different light conditions showed a dose-dependence effect of melatonin to inhibit the adhesion of the integrin-mediated granulocyte to intercellular adhesion molecule-coated plates via melatonin receptors [136]. Taken together, the pleiotropic role of melatonin would contribute to the effect of downregulation toward adhesion and invasion.

### 3.5. Anti-Epithelial–Mesenchymal Transition (Anti-EMT)

As EMT is defined as the process whereby epithelial cells are transformed into mesenchymal cells, it is an essential process to regulate endometriosis progression and metastasis [137]. To undergo the process of EMT, cells are required in the cycle of migration, invasion, and proliferation. In addition, signaling pathways, such as NF-κB, Wnt, Notch, and other crucial factors (e.g., Snail, Slug, and Twist), commonly regulate the EMT process via modulating the expression levels of E-cadherin, N-cadherin, and vimentin [138,139,140,141], such that the Notch signaling pathway has been shown to be critical for the induction of EMT and the development of various diseases, including endometriosis [142,143]. Melatonin has been shown to inhibit Estradiol-17β-induced EMT in normal epithelial cells and endometriotic cells by increasing the expression of Numb and decreasing activity of the Notch signaling pathway [64]. Besides its anti-EMT effect on endometriosis, melatonin inhibited the EMT process and tumor progression in human gastric cancer cell lines by upregulating E-cadherin expression via the NF-κB signaling pathway [144,145]. A reduction in vimentin expression was equally observed when the rates of migration and invasion of mammary spheres formed by canine and human breast cancer cell lines were suppressed by NF-κB signaling after melatonin treatment [146]. Likewise, the Wnt/β-catenin pathway is another important regulator of EMT for β-catenin to interact with E-cadherin, which acts as a key factor for adhesive binding [139]. In a similar matter, melatonin can activate GSK3β via upregulating the Akt phosphorylation level to stimulate β-catenin metabolism, which ultimately inhibits EMT [147]. In addition, melatonin was found to suppress RSK2 via the HER2/MAPK/Erk pathway to inhibit EMT in breast cancer cells [148]. Besides these pathways, melatonin has also been shown to inhibit transforming growth factor beta-induced EMT in MG-63 osteosarcoma cells via the HIF-1/Snail/MMP-9 pathway [149].

### 3.6. Anti-Angiogenesis

Angiogenesis serves as a critical function during the development of endometriosis for facilitating the formation of new blood vessels to circulate nutrients and oxygen and to stimulate the progression of endometriotic lesions [150]. Activities of angiogenic factors, in particular vascular endothelial growth factor (VEGF), were higher in both peritoneal fluid and endometrium in women with endometriosis than controls [151,152]. The stimulation by VEGF can promote proliferation, migration, tubular organization, and increased permeability of endothelial cells, ultimately favoring angiogenesis [153]. Based a meta-analysis evaluating the treatment with inhibitors of VEGF/VEGFR agents in animal models of endometriosis, the capability to reduce the endometriotic lesion sizes without influencing the number of follicles was shown [154]. Similarly, a surgically induced endometriosis animal model demonstrated that suppression of the isoform of VEGF, notably VEGFC, can also suppress angiogenesis and the growth of endometriotic lesions [155,156]. In another study, melatonin was found to decrease the VEGF expression level for anti-angiogenic activity for reducing implant volumes in an endometriosis model [51]. Additionally, melatonin was also shown to regulate VEGF and VEGFR expression in physiological and pathological situations. In a breast cancer xenograft model, melatonin suppressed the expression of VEGFR-2, the most critical receptor of VEGF, to significantly reduce vascular density and angiogenesis [157], while in HUVECs, melatonin showed a dose-dependent inhibition of VEGFR-2 induced by VEGF [158]. In addition, oral administration of melatonin decreased serum VEGF levels in patients with metastatic cancer [159]. Additionally, in MCF-7 cells, melatonin treatment reduced VEGF-C and VEGF receptors (VEGFR2 and VEGFR3) during hypoxia [134]. Several in vitro cancer studies demonstrated that melatonin interfered with hypoxia-inducible factor-1a (HIF-1α), which is the transcription factor and an upstream of VEGF. In a mouse model of renal cancer, the effect of melatonin on the expression of HIF-1α and VEGF was determined to be dependent on its antioxidant activity [159]. Likewise, melatonin reduced the growth of ectopic uterine tissue in an endometriosis rat model through the modulation of VEGF [51]. Furthermore, two vascular growth factors, angiopoietin-1 (Ang-1) and Ang-2, were noted to play crucial roles in angiogenesis. Their expressions in the eutopic endometrium of women with endometriosis were found to be increased when compared to normal uterine endometrium [160]. Melatonin lowered levels of ANG-1, ANG-2, and VEGF in the co-culture medium and their respective mRNA expressions in HUVEC and MCF-7 cell lines, respectively [45]. Unlike other angiogenic inhibitors solely targeting specific molecules, melatonin would act to alter several important angiogenic factors and processes to suppress vascularization in various diseases.

### 3.7. Immune Modulation

The immune system plays an important role in the pathophysiology and symptomatology of endometriosis [161,162]. Both the peripheral immune system and the eutopic endometrial immune situation were altered in women with endometriosis [163,164]. Unlike healthy individuals, macrophages (Mø), immature dendritic cells (DC), and regulatory T cells displayed different behaviors in women with endometriosis [165]. Besides immune cells, cytokines would also be altered in women suffering from endometriosis. Such cytokines as interleukin (IL) 6, IL-8, and cancer antigen (CA) 125 were shown to be upregulated, while tumor necrosis factor (TNF) α was found to be downregulated in patients with endometriosis. These alternations in the immune system would evidently reduce the immunological surveillance in endometriosis. Hence, the determination of these cytokines can often act as a diagnostic index of endometriosis [166,167]. In a mice study, the numbers of monocytes and natural killer (NK) cells were increased in those treated with either 7 or 14 days of melatonin [168]. Melatonin also enhanced antigen presentation via interaction between macrophages and T-lymphocytes, which can lead to the activation and proliferation of cytotoxic T lymphocytes [169]. In tumor-bearing rats, melatonin also showed its stimulatory potential on lymphocytes, monocytes/macrophages, and NK cells [170]. Recently, a meta-analysis analyzed 31 randomized and non-randomized placebo-controlled clinical trials with a total sample size of 1517 participants. It showed that melatonin supplementation significantly decreased TNF-α and IL-6 levels [171]. Taken together, melatonin has a direct immune regulatory effect in both animals and human studies.

### 3.8. Neurotrophic Effect

Pain is the most common symptom resulting from endometriosis. Endometriotic lesions cause chronic pelvic pain via compressing or infiltrating the nerves located close to the lesions [172]. Although the underlying mechanisms are elusive, emerging evidence has shown that endometriotic lesions and peritoneal fluid in women with endometriosis exhibit outstanding neurotrophic properties, including the increased expression of new nerve fibers, and the upregulation of various neurotrophies [173]. Brain-derived neurotrophic factor (BDNF), or abrineurin, is a member of the neurotrophic class of growth factors. This is a key regulator of neuronal survival and neurogenesis [174]. As reported, BDNF has the capacity to modulate pain pathways at different levels, from the peripheral nociceptors to spinal cord neurons and the brain [175]. A phase II clinical trial demonstrated that melatonin downregulated the secretion of BDNF to reduce pain scores as indexed by the visual analog scale (VAS) that includes daily pain, dysmenorrhea, dyspareunia, dysuria, and dyschezia. In addition, this study showed that melatonin treatment could be associated with an improved sleeping quality in patients with endometriosis [18]. Typically, melatonin has been suggested to be a well-tolerated alternative treatment for the pain-associated symptoms of endometriosis. In addition, melatonin possesses neuroprotective functions and is often considered as a potential neurological drug in clinics, such that treatment with melatonin in newborn piglets increased hypothermic neuroprotection by improving brain energy metabolism and reducing brain damage [176]. It reduced the expression of astrogliosis, whose accumulation would trigger the production of pro-inflammatory cytokines leading to secondary damage [177]. In rat experimental models of neuropathic conditions, which include Alzheimer’s disease, Parkinson’s disease, and ischemic stroke, melatonin showed its effectiveness in arresting neurodegenerative phenomena [178]. Furthermore, during the dark phase, it appeared that mice with a higher melatonin level in plasma were less susceptible to a nociceptive stimulus [179]. The analgesic effects of melatonin were undeniable and evident, which had been documented in various animal models and clinical trials of patients with different pain syndromes. Animals treated with melatonin demonstrated its antinociceptive effect, thus alleviating the hypersensitivity induced in the hot plate test, electrical stimulation, and nerve ligation [180,181,182]. In clinical studies, melatonin reduced the frequency and intensity of pain in migraines, cluster headaches, tension headaches, chronic back pain, and fibromyalgia [183,184,185,186,187]. Considering the therapeutic application of melatonin from both its neuroprotective and antinociceptive functions, melatonin should be considered as a potential drug for treating endometriosis-associated pain.

## 4. Clinical Insights of Melatonin in Endometriosis and Other Reproductive Diseases

Currently, there are many ongoing clinical studies on using melatonin as a therapeutic strategy, including insomnia, diabetes, fibromyalgia, hypertension, anticancer, and aging-related diseases, such as Parkinson’s and Alzheimer’s disease [188,189,190,191]. In the obstetrics and gynecology discipline, there was only one randomized controlled trial in melatonin for endometriosis in 2013. For other reproductive disorders, melatonin has shown its clinical therapeutic values in various aspects (Table 2). Based on a clinical trial, it was shown that melatonin supplements could relieve pelvic pain and improve the sleeping quality of women with endometriosis [18]. During application for IVF-ET with infertile patients, the administration of melatonin can improve the oocyte quality [192]. The number of degenerated oocytes was significantly reduced in the 3 mg melatonin group when compared to those in the control group [193]. Additionally, the number of fertilized embryos was also shown to have an increased tendency in the 3 mg melatonin group [194]. Likewise, intrafollicular melatonin concentrations were slightly increased in the 1 mg melatonin group, and concentrations were further increased in the 3 mg melatonin group. After treatment of melatonin, the concentration of intrafollicular lipid peroxide, a free-radical reaction that causes injury to oocytes, showed a tendency towards a lower level, the level of significance being dependent on melatonin dosage [195]. Melatonin increased follicular concentrations of it, thereby inhibiting oxidative stress and improving the quality of oocytes [195]. A further study also examined the possibility of whether melatonin therapy inhibited oxidative stress to improve the quality of oocytes and the clinical outcome of IVF-ET [196]. The result in this study reported improved concurrent fertilization and pregnancy rates with melatonin treatment [196]. These initial studies inspired other researchers to examine whether the use of melatonin can help to increase the number of mature oocytes for improving the fertilization rate and the clinical outcome of IVF-ET. In general, melatonin supplementation was shown to increase the number and quality of both oocytes and embryos. Moreover, it could increase the clinical pregnancy rates of patients with infertility disorders or low fertility rates [47,192,195,197]. As poor oocyte quality is a major cause of infertility in endometriosis, whether melatonin can increase the fertility rate through improving oocyte qualities requires further studies. In addition, treatment with melatonin was able to restore menstrual cyclicity in women with polycystic ovarian syndrome (PCOS) [198]. Daily supplementation of melatonin in PCOS women showed a significant improvement in pregnancy rates [199]. Importantly, the overall effects of melatonin supplementation in fetal growth restriction, pre-eclampsia, and climacteric symptoms appear to be promising and clinically relevant [200,201,202]. Likewise, women with endometriosis also have menstrual and poor obstetrics outcomes as summarized in Table 2.

Therapeutic strategies that are safe and with minimal reproductive side effects are essential to women with endometriosis. Melatonin is believed to be a potential candidate as a safe supplementation toward treating endometriosis and endometriosis-associated symptoms [210]. Owing to the limited clinical trials of melatonin in endometriosis, the other clinical trials utilizing melatonin as a potential therapeutic solution on other reproductive diseases may help to support its application toward endometriosis.

## 5. Conclusions

Importantly, this review has helped to indicate the potential therapeutic capabilities of melatonin for the treatment of endometriosis and its associated symptoms. This is attributed to its many properties, including antioxidation and anti-inflammation, its ability to modulate the endocrine functions via hormonal signaling pathways, and the absence of toxic side effects. Additionally, there is accumulating evidence which illustrates the application of exogenous melatonin to suppress ectopic endometriotic lesions, relieve endometriosis-associated pelvic pain, and enhance the sleeping quality of women with endometriosis. Nevertheless, further analysis on determining the melatonin levels in the eutopic and ectopic endometrium of women with endometriosis for therapeutic efficacy is still warranted. Moreover, corroborating evidence from mechanistic studies and randomized controlled trials would be necessary for better understanding the pharmakinetics and the clinical application of melatonin for endometriosis treatment.

## Figures and Tables

**Figure 1 nutrients-14-04087-f001:**
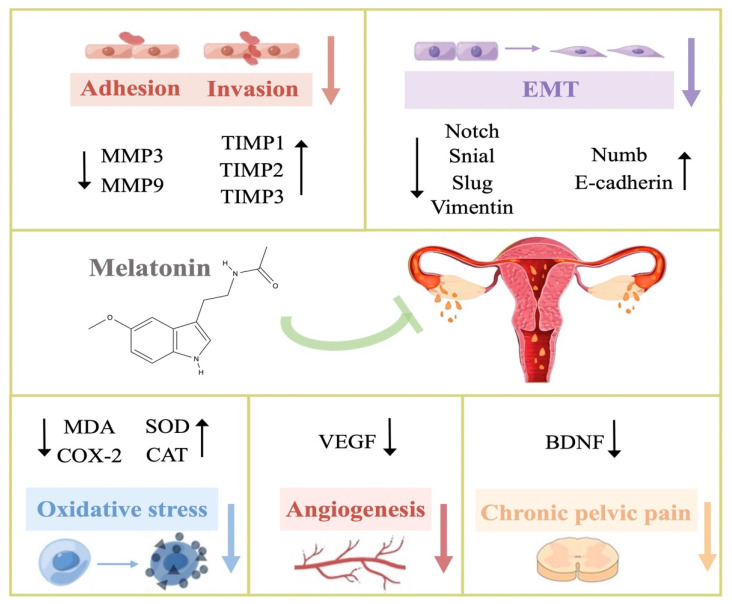
Therapeutic targets of melatonin toward inhibiting the development and progression of endometriosis. The effect of melatonin with respect to various molecular targets involved in the processes of endometriosis, including adhesion, invasion, EMT, oxidative stress, angiogenesis, and chronic pelvic pain. Abbreviations: MMP, Matrix metalloproteinase; TIMP, Tissue inhibitors of metalloprotease; EMT, Epithelial–mesenchymal transition; MDA, Malondialdehyde; COX-2, Cyclooxygenase 2; SOD, Superoxide dismutase; CAT, Catalase; VEGF, Vascular endothelial growth factor; BDNF, Brain-derived neurotrophic factor.

**Table 2 nutrients-14-04087-t002:** Clinical trials of melatonin for endometriosis and other female reproduction.

Studies	Disorder	Sample Size (Melatonin/ Control)	Age (Means ± SD/ SEM or Range)	Daily Dose	Duration or Period	Main Outcomes	Adverse Events	Trial Registration Number
Schwertner 2013 [18]	Endometriosis	20/20	18–45	10 mg	8 weeks	Reduced pain scores and BDNF levels Increased sleep quality	/	/
Takasaki 2003 [195]	Infertility	27/NA	34 ± 4.4	1 or 3 mg	From 2nd to 5th day of the menstrual cycle to hCG day	Decreased number of degenerate oocytes (only in 3 mg group)	/	/
Tamura 2008 [47]	Infertility	56/59	34.8 ± 4.8	3 mg	From 2nd to 5th day of the menstrual cycle to OPU day	Increased oocytes quality and fertility rate	/	/
Rizzo 2010 [203]	Infertility	32/33	35–42	3 mg	From GnRH administration day to pregnancy test result was confirmed	Increased tendency on number of mature oocytes, embryo quality, clinical pregnancy rate, and implantation rate	/	/
Batioglu 2012 [193]	Infertility	40/45	20–40	3 mg	During the IVF-ET procedure	Increased number and percentage of mature oocytes Increased number of embryos of top quality Increased tendency on clinical pregnancy rate	/	/
Nishihara 2014 [204]	Infertility	97/NA	≤42	3 mg	At least 2 weeks	Increased fertilization and embryo quality	/	/
Fernando 2018 [205]	Infertility	32/29 29/29 26/29	18–45	2, 4, or 8 mg	From 2nd to 5th day of the menstrual cycle to OPU day	No significant differences in daytime Karolinska sleepiness score between groups No differences in objective measures of sleep quantity or quality Improved the subjective sleep quality scores except 8 mg group	/	ACTRN12613001317785
Fernando 2018 [194]	Infertility	41/41 39/41 40/41	18–45	2, 4, or 8 mg	From 2nd to 5th day of the menstrual cycle to OPU day	There were no differences between all the groups in total oocyte number, number of MII oocytes, number of fertilized oocytes, the number or quality of embryos, clinical pregnancy rate, or live birth rate	Headache, fatigue (no significant difference)	ACTRN12613001317785
Espino 2019 [196]	Infertility	20/20		3 or 6 mg	From 2nd to 5th day of the menstrual cycle to OPU day	Ameliorated intrafollicular oxidative balance and oocyte quality Increased the rate of pregnancies/live births No significant difference between 3 mg group and 6 mg group	/	/
Wdowiak 2020 [206]	Infertility	50/50	20–35	1 mg	6 months	Increased blastocyst and oocyte quality Increased rate of clinical pregnancy	/	/
Jahromi 2017 [207]	Diminished ovarian reserve	40/40	22–42	3 mg	From 2nd to 5th day of the menstrual cycle to OPU day	Increased serum estradiol level on the triggering day Increased number of MII oocytes, top quality embryos with G1 and G2	/	IRCT2014041417264N1
Tagliaferri 2017 [198]	PCOS	40/NA	23.25 ± 4.07	2 mg	6 months	Improved the menstrual irregularities Decreased biochemical hyperandrogenism	/	/
Mokhtari 2019 [199]	PCOS	98/100	28.9 ± 5.5	3 mg	From 2nd to 5th day of the menstrual cycle to hCG day	Increased chemical pregnancies and endometrial thickness	/	/
Parandavar 2018 [208]	Menopausal women	98/142	53.22 ± 4.21	3 mg	3 months	Increased amount of triglyceride	/	/
Chojnacki 2018 [209]	Postmenopausal women	30/30	51–64	8 mg	12 months	Decreased Kupperman Index and body mass index	No adverse influence on patients’ psychosomatic activity	/
Miller 2014 [200]	Fetal growth restriction	6/6	/	8 mg	For the duration of pregnancy	Decreased placental oxidative stress	No adverse maternal or fetal effects	NCT01695070
Parandavar 2014 [202]	Climacteric symptoms	99/101	40–60	3 mg	3 months	Decreased climacteric symptoms score	Sleepiness, nausea, vomiting, and headache (no significant difference)	/
Hobson 2018 [201]	Preeclampsia	20/48	Control: 30.6 ± 0.8 Melatonin: 32.7 ± 1.1	30 mg	From recruitment until delivery	No adverse events and adverse drug reactions Prolonged gestation Reduced the pharmacological need for antihypertensives	No adverse events	/

ACTRN, Australian clinical trials registration number; BDNF, Brain-derived neurotrophic factor; hCG, Human chorionic gonadotropin; GnRH, Gonadotropin-releasing hormone; IRCT, Iranian Registry of Clinical Trials; IVF-ET, In vitro fertilization and embryo transfer; MII, Metaphase II; NA, Not applicable; NCT, National clinical trial; OPU, Oocyte pick up; PCOS, Polycystic ovary syndrome; SD, Standard deviation; SEM, Standard error of mean; SOD, Superoxide dismutase.

## Data Availability

Not applicable.

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
