# Peer review of "Melatonin in Endometriosis: Mechanistic Understanding and Clinical Insight"

_nutrients, 2022, doi:10.3390/nu14194087_

Round 1
Reviewer 1 Report
Review of the manuscript nutrients-189223, entitled "Melatonin in endometriosis; mechanistic understanding and clinical insight." by Dr. Li et al.
Overview and general coments
This article was conducted to understand the biology and molecular mechanism of melatonin and emphasizes its potential clinical applications to treat endometriosis.
The involvement of melatonin in endometriosis pathophysiology seems to be valuable and clinical application of melatonin on the patients with endometriosis is important issue.
Major coments
1. The articles of melatonin related to endometriosis are limited to studies using animal model, and most of its mechanisms of action are due to its anti-oxidative and anti-apoptotic effects. Other mechanisms of action of melatonin on endometriosis pathophysiology are speculative without sufficient evidence.
2. There is only one clinical trial that showed melatonin supplements could relieve pelvic pain and improved the sleeping quality of women with endometriosis. So far, there is no clinical trial of evaluating the therapeutic ability of melatonin on endometriotic lesions. It is premature to write a clinical review.
3. There is no author’s article in the references. The authors have not published any articles on endometriosis or melatonin research. The authors’ contribution to endometriosis and melatonin research is unclear.
Reviewer 2 Report
The manuscript entitled “Melatonin in endometriosis mechanistic understanding and clinical insight” and authored by Li and colleagues, deals with the latest in vitro and in vivo findings toward evaluating the pleiotropic functions of melatonin, as well as to summarize its physiological and pathological effects and treatment capabilities in endometriosis.
The topic of the article is really interesting. Melatonin is a metabolite whose functions have not yet been fully elucidated. The content is really of high quality, and the manuscript is really well written, although some typos are present in the main text. I would strongly recommend rereading the manuscript, and fix them as much as possible before resubmitting it. However, I do not see this as a reason not to achieve its publication in Nutrients.
In addition, I feel compelled to suggest some substantial changes to the authors to improve the quality of the manuscript.
(i) The abstract should be more comprehensive, better explaining the main issues addressed by the authors in the review.
(ii) Regarding keywords, both should be seriously changed, and new ones added. Authors should follow the directions of the paper, and provide keywords (maximum 10) that are inherent to the purpose of the manuscript, and that are not contained in the title, or partially contained in the abstract. This is very important point. The usefulness of keywords is to make the article more visible after publication. Other authors may be more easily directed to your manuscript when search engines are used. That is, if the right keywords are chosen.
(iii) The authors should check the references well. For example, some seem to be out of place. They should also standardize the formatting. Preferably, it would be better to use ISO 4 rules. Line 51 refers to a reference 1010. I think it is a typos.
(iv) Synthesis (section 2.1.), metabolism (2.2.) and receptors (2.3.) is not as simple and reductive as reported by the authors. The authors should refer to this recent manuscript, where the mechanism of synthesis and metabolism is clearly reported in animals (10.3390/ijms22189996). In particular, the authors should introduce more precise information regarding the effects of melatonin in animals.
(v) Table 1 and Table 2 lend themselves very well to a forest plot analysis. Through this analysis, the authors could obtain a chart in which potential beneficial effects are shown through a graph, comparing with control groups. This is not a mandatory point, although it could seriously implement the quality and originality of the manuscript.
Round 2
Reviewer 1 Report
I rejected this article.
Author Response
Dear reviewer,
We would like to thank you once again for the comments and suggestions to improve our manuscript.
Sincerely and on behalf of all authors,
Yiran LI,
PhD student
Department of Obstetrics & Gynaecology
The Chinese University of Hong Kong